# Phenotypic and genotypic detection of extended spectrum beta lactamase enzyme in *Klebsiella pneumoniae*

**Aso Bakr Mohammed**[1], **Khanda Abdullateef Anwar**[2]*

1 Microbiology Department, Shar Teaching Hospital, Sulaimani, Iraq, 2 Microbiology Department, College of Medicine, University of Sulaimani, Sulaimani, Iraq

* Khanda.anwar@univsul.edu.iq

## Abstract

### Background

*Klebsiella* species are ubiquitous in nature and can be found in the natural environment and on mucosal surfaces of mammals and it is an important multidrug-resistant pathogen affecting humans and is a major source for hospital acquired infections. The aim of this study is to investigate the prevalence of ESBL enzyme among *Klebsiella pneumoniae* isolates by phenotypic methods from different hospital wards and detection of ESBL resistance genes such as TEM and SHV in Sulaimani city/ Kurdistan–Iraq.

### Methods

*Klebsiella pneumoniae* isolates were collected from different clinical samples from different hospitals, the isolates were identified by standard technique. Screening of ESBLs was undertaken by using double disk diffusion and standard disk diffusion methods. Real-time PCR was used for genotypic detection of TEM and SHV genes according to the standard protocol.

### Result

Out of 54 *Klebsiella pneumoniae* isolates; 28 were ESBL positive, The pattern of antimicrobial susceptibility testing showed that the most resistant antibiotic are AMP (100%), AMC (100%) followed by CAZ (83.33%), CTX (75.9%), CPM (74%), ATM (70.37%). Both TEM and SHV genes were detected among 28 (51.85%) ESBL positive by using Real-time PCR method.

### Conclusion

SHV gene was detected in most of the isolates of ESBL producers of *Klebsiella pneumoniae*.

**Data Availability Statement:** All relevant data are within the manuscript and its Supporting information files.

**Funding:** The author(s) received no specific funding for this work.

**Competing interests:** The authors have declared that no competing interests exist.

## Introduction

Extended-spectrum beta-lactamase (ESBL) one of the most common types of antibiotic resistance found in *Enterobacteriaceae*, plasmid-mediated enzymes capable of hydrolyzing a β-lactam bond in commonly prescribed β-lactam antibiotics, such as penicillins, broad spectrum cephalosporins and aztreonam [1] these enzymes have been clustered into nine different structural and evolutionary families based on their amino acid sequence.

TEM and sulphydryl variable SHV were the major types however, CTX-M type is more common in some countries [2] and they are rapidly evolving and produced by Gram negative bacteria, and have the ability to hydrolyse all cephalosporins, aztreonam, and related oxyimino-beta lactams as well as older penicillins but this can be inhibited by clavulanic acid [3].

The plasmids that harbor ESBL genes also carry genes for other antibiotic classes such as aminoglycosides, chloramphenicol, sulfonamides, trimethoprim, and tetracycline. Thus, Gram negative bacilli containing these plasmids mostly are MDR bacteria [4].

A large number of outbreaks of infections due to ESBL producing organisms have been described all over the world. In some hospitals, initial outbreaks of infections have been supplanted by endemicity of the ESBL producing strains [5]. This may lead to increased patient mortality, therefore, control of the initial outbreak of ESBL producing organisms in a hospital or specialized unit of a hospital is of critical importance [6].

About 400 different types of ESBLs have been recognized around the world, among which TEM and SHV were more prevalent, mutations occurring in the genes encoding TEM and SHV enzymes lead to the development of ESBLs that have an expanded substrate profile [7].

Both TEM and SHV types are derived from penicillinase that is characterized by several single amino-acid substitutions, these mutations allow them to hydrolyze extended spectrum cephalosporins [8].

TEM-1 is the most commonly encountered β-lactamase in Gram-negative bacteria. Up to 90% of ampicillin resistance in *E. coli* is due to the production of this enzyme, and it is also responsible for the ampicillin and penicillin resistance that is seen in *H. influenzae* and *N. gonorrhoeae* [9].

The fact that, patients most likely to become infected with ESBL-producing Enterobacteriaceae are those with prolonged stays in the intensive care unit (ICU). Failure to rapidly detect ESBL-producing strains may result in serious consequences, such as treatment failure and death [10].

It is estimated that *Klebsiella pneumoniae* cause 8% of all hospital-acquired infections. In the USA it comprise 3% to 7% of all nosocomial bacterial infections, this place them among the eight most common pathogens in hospitals and second only to *E. coli* as the most common cause of Gram-negative sepsis [11], while urinary tract is the most common site for infection of *Klebsiella* species and it accounts for 6 to 17% of all nosocomial urinary tract infections (UTI) and shows an even higher incidence in specific groups of patients at risk, such as patients with neuropathic bladders or with diabetes mellitus [12].

The likelihood of *Klebsiella pneumoniae* hospital-acquired infections is greatly increased by the presence of invasive devices, such as catheters, and ventilator associated pneumoniae in hospitalized patients [13].

Antimicrobial resistance in *Klebsiella pneumoniae* has been attributed largely to the acquisition of large, self-conjugating plasmids. These are extra chromosomal DNA molecules that encode factors promoting the initiation of replication, independently from the replication of the bacterial chromosome, maintain a constant copy number per bacterial cell and regulate their own replication rate [14].

Carriage of multiple β-lactamase genes in the same strain is a known ability of *Klebsiella pneumoniae* and may contribute to the selective success of this pathogen [15]. Combinations of all types of *bla* genes were reported in this species, this may be due to either carriage of an antibiotic-resistant plasmid encoding an array of antibiotic resistance genes or due to acquisition of transposons containing different *bla* genes on the same plasmid [10].

Many different techniques exist for confirming ESBL production, Phenotypic tests for ESBL detection only confirm whether an ESBL is produced but cannot detect the ESBL subtype. Some ESBLs may fail to reach a level to be detected by disk diffusion tests but result in treatment failure in the infected patient [16], while those utilizing similar methodology to standardized susceptibility tests are the most convenient for the routine diagnostic laboratory. These all depend on detecting synergy between clavulanic acid and the indicator cephalosporin used in the primary screening [17].

Another confirming method involves at least an eight-fold reduction in the MICs of ceftazidime or cefotaxime in the presence of clavulanic acid, the isolates then reported as resistant to all penicillins, cephalosporins and aztreonam [8].

Early detection of β-lactamase genes was performed using DNA probes those were specific for TEM and SHV genes. However, using DNA probes can sometimes be rather labor intensive. The easiest and most common molecular method used to detect the presence of a β-lactamase belonging to a family of enzymes is PCR with oligonucleotide primers that is specific for a β-lactamase gene [18].

Molecular methods appears to be sensitive in spite to that its expensive, time consuming and require specialized equipment and expertise [19].

Determination of TEM and SHV genes by molecular techniques in ESBL producing bacteria and their pattern of antimicrobial resistance can supply useful data about their epidemiology and risk factors associated with these infections [20].

## Methods

This cross sectional study was carried out in different hospitals (Shar teaching hospital, and Dr. Jamal Ahmed Rashid hospital) in Sulaimani city from a period of March to July 2019. Ethical approval was obtained from Directorate of Health of Sulaimani.

Fifty-four isolates of *Klebsiella pneumoniae* were collected from all the mentioned hospitals from different infection sites and the samples underwent processing through standard bacteriological technique by doing Gram stain and culture technique for all the isolates [21].

All isolated *Klebsiella pneumoniae* were tested by string test through formation of a viscous filament ≥5 mm after stretching a bacterial single colony (Fig 1) with a standard bacteriological loop on an agar plate [22, 23] VITEK 2 bioMerieux GN76 / ID card machine was used for the identification of all isolated *Klebsiella* according to the manufacturer's recommendations. Bacterial strains which were identified biochemically were preserved for longer period by transferring a fresh colony from newly culture plates to prepare 0.5 mL bacterial suspension in tryptic soya broth and they incubated for 18 hours at 37˚C in shaking incubator, after which 0.5 mL of this growth cultures was mixed with 0.5 mL of sterile 80% glycerol solution in a Eppendorf tube. After labeling the tubes with numbers and dates of preservation and stored in a deep freezer at -70˚C [24].

Antibiotic susceptibility pattern of *Klebsiella pneumoniae* isolates were performed according to the Kirby–Bauer discs diffusion method [25] by using 15 commonly used antibiotic agents; Ampicillin (AMP), Amoxicillin-Clavulanic Acid (AMC), Piperacillin-tazobactam (PTZ), Cefoxitin (FOX), Ceftazidime (CAZ), Ceftriaxone (CRO), Cefotaxime (CTX), Cefepime (CFM), Aztronam (ATM), Imipenem (IPM), Meropenem (MEM), Amikacin (AN),

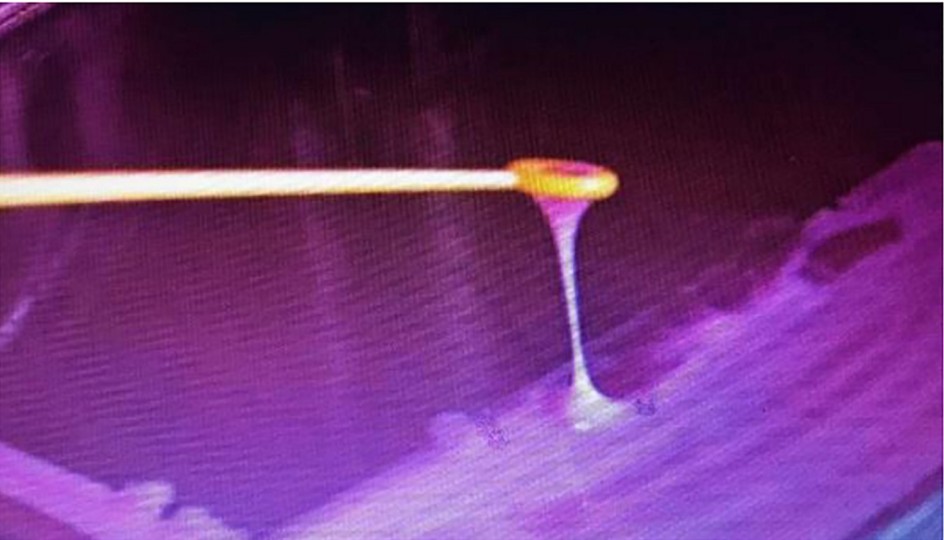

**Fig 1. String test: A hypermucoviscous phenotype is seen when a viscous filament ≥ 5 mm is produced after stretching of *Klebsiella pneumoniae* colony with a loop on agar plate.**

Gentamicin (CN), Ciprofloxacin (CIP), Trimethoprim- Sulphomethoxazole (TMP), and all of the inoculated plates were aerobically incubated at 37˚C for 18–24 hours, and the results were interpreted based on the instruction provided by the CLSI [26].

## Detection of extended spectrum β-lactamase enzyme

All isolates of *Klebsiella pneumoniae* were screened for ESBL enzyme production by using various antimicrobial disks, ESBL positive meant the bacteria shows high level (co-resistance) to third generation cephalosporin and monobactams such as: Ceftazidime zone ≤ 22 mm, Aztreonam zone ≤ 27 mm, Cefotaxime zone or Ceftriaxone zone ≤ 25 mm [26].

**Double disk synergy test (DDST).** The double-disk synergy method is integrated as an adjunct of the routine susceptibility test. In this test, a disk of amoxicillin-clavulanic acid AMC (30μg) and disks of the third generation Cephalosporin antibiotic such as Ceftazidime CAZ disk (30μg), Cefotaxime CTX (30μg), Ceftriaxone CRO (30μg), and Cefepime FEP (10μg), were used together with Aztronam ATM (30μg), each disks were placed at a distance of 20 mm apart from the inhibitor disk on a lawn culture of the resistant isolate on Mueller-Hinton Agar plate. The presence of ESBL was inferred when the inhibition zone around any of the antibiotic disks was enhanced on the side of the Clavulanic acid-containing disk (Fig 2), resulting in a characteristically shaped zone referred to as a "keyhole [27].

**Detection of ESBL by VITEK 2 compact system.** Each isolate was tested using the **VITEK 2 system** with the antimicrobial susceptibility test extend AST-EXN8 card. This system was designed to perform both screening and confirmatory tests for phenotypic detection of ESBL on the same plate [28].

## Genotypic detection of ESBL enzyme

Two genes (*bla*SHV and *bla*TEM) were selected in this study to be analyzed by real time PCR. Bacterial reactivation was performed by thawing frozen samples (bacterial culture and glycerol), and sub cultured on blood agar then incubated for 24 hours, selected colony from blood

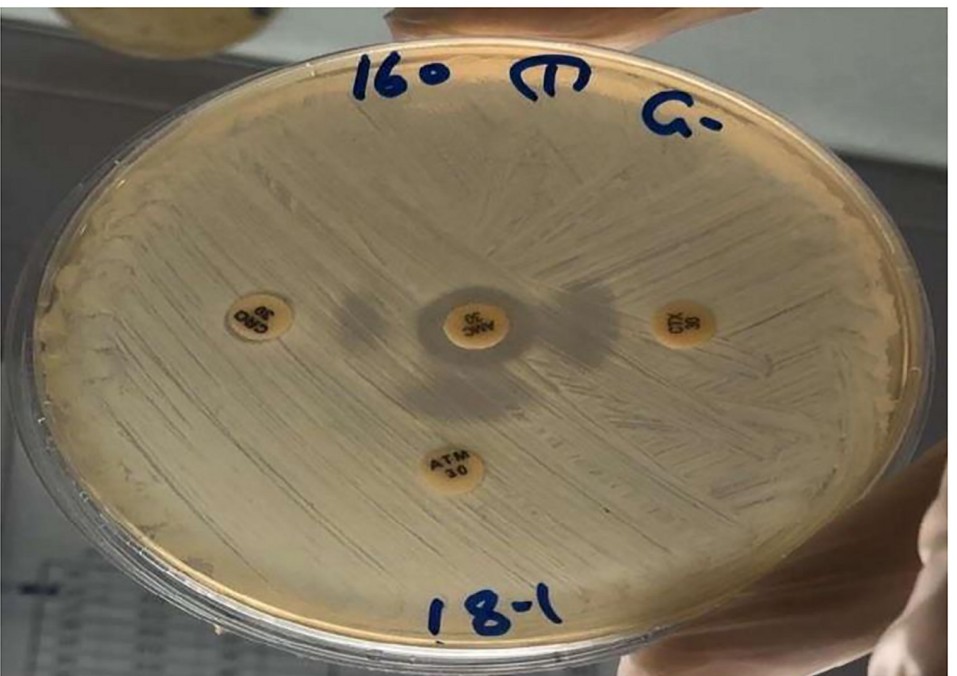

**Fig 2. Double disc synergy test for ESBL.**

agar was sub cultured in to Tryptic soy broth for 18 hours incubated at 37°C, and samples were ready for plasmid extraction [29].

**Primer preparation.** Two sets of primers were used in this study and the primers were designed by Macrogen/Korea both forward and revers primer were designed previously "Table 1" [30].

Primers were prepared according to what was recommended by manufacturer (Macrogen) by dissolving lyophilized primer in 300 μL of DNA and RNA injection water to final concentration of stock primers at 100 pmol/μL. and dilution of 1:10 (10 μL of stock primer with 90 μL of DNA and RNA free distil water) were accepted to prepare a final working primer in concentration 10 pmol/μL [31].

Plasmid extraction was performed as recommended by the manufacturer of (MagCore®, RBC bioscience) automated nucleic acid extraction kit (Taiwan), prior to plasmid extraction, the content of the kit was prepared and checked as recommended.

**SYBR green real-time PCR for *bla*TEM and *bla*SHV detection.** Real-time PCR was used to detect of TEM and SHV genes, PCR was performed by rapid cycling in a reaction volume 20 μL including (master mix, amplification primers, extracted nucleic acid, and DNA and RNA free distil water). All isolates with ESBL positive by phenotypic methods were underwent

**Table 1. *bla* SHV and *bla* TEM primer sequence.**

| Target gene | Nucleotide | Amplicon size (bp) | Reference |
|---|---|---|---|
| bla TEM gene -F | 5-TCCGCTATGAGACAATAACC-3 | 296 | Doosti et al., 2015 |
| bla TEM gene -R | 3-ATAATACCGCACCACATAGCAG-5 | | |
| bla SHV gene -F | 5-TACCATGAGCGATAACAGCG-3 | 450 | |
| bla SHV gene -R | 3-GATTTGCTGATTTCGCTCGG-5 | | |

**Table 2. Programs of real-time PCR thermocycling condition after serial optimization.**

| Gene | Initial dentauration ˚C/time | Denaturation | Annealing | Extension | Number of cycle | Final extention |
|---|---|---|---|---|---|---|
| *bla* HSV | 95˚C/5 minute | 94/ ˚C 40 sec | 59/ ˚C 40 sec | 72/ ˚C 40 sec | 35 | 72/ ˚C 40 sec |
| *Bla* TEM | 95˚C/5 minute | 94/ ˚C 40 sec | 59/ ˚C 40 sec | 72/ ˚C 40 sec | 30 | 72/ ˚C 40 sec |

A negative control was included in each run to access specificity of primers and possible contamination. The threshold cycle (Ct) was determined as the PCR cycle at which increased reporter fluorescence above the baseline could be detected.

All samples were run on PCR machine, PCR reaction data were plotted on graphic presentation by Micro software PCR program.

plasmid extraction by automated nucleic acid extractor, 7.5 μL of extracted nucleic acid in concentration 10 ng/μL mixed with 10 μL of qPCR master mix and 1.5 of DNA and RNA deionized water, 0.5μl of forward primer and 0.5 μl of reverse primer at concentration 10 ng/μl then added in to the reaction mic PCR tube and the condition of PCR, after that the reaction tube subjected the thermo cycling reaction on bio molecular system (BMS) model instrument (Australia), according PCR condition used in amplification cycle "Table 2".

*Statistical analysis.* All data were analyzed by transfer to excel sheet data, by using Microsoft Excel 2010 and *P* value < 0.05 was regarded as significant.

## Results

Fifty four suspected colonies of *Klebsiella pneumonia* were collected from different hospital laboratories in Sulaimani city, out of 54 samples; 29 (53.7%) of all specimen were males and 25 (46.3%) female, with male to female ratio 1.15:1.

All colonies of 54 samples showed pink color, moist and mucoid colony on the surface of MacConkey agar and large grey color colony on blood agar with no hemolytic property among all isolated bacteria. Out of the 54 isolated *Klebsiella pneumonia*, only 8 samples were positive to hypermucoviscosity phenomena according to what was described previously [32].

Out of 54 specimens which had been received; the highest numbers were taken from intensive care unit 32, (59.26%), followed by dialysis center 5 (9.26%), 3 (5.55%) for each of Pediatric and outpatient, and other wards in hospitals 11 (20.37%) that is illustrated in "Table 3". The result indicate that among all specimens taken from all hospital wards; urine were 23 (42.6%), followed by Endotracheal aspiration 12 (22.22%), blood 11 (20.37%), catheter tip 4 (7.41), wound swab 3 (5.55%), and 1 (1.85%) was recorded for sputum.

"Table 2" All confirmed samples were tested for antimicrobial susceptibility testing, by using Kirby-Bauer disc diffusion method and the result were analyzed according to CLSI [21].

**Table 3. Clinical specimens and source of samples of *Klebsiella pneumoniae*.**

| Specimen | ICU | Dialysis | Pediatric | OPD | Others | Total |
|---|---|---|---|---|---|---|
| Urine | 9 | 1 | 2 | 3 | 8 | 23(42.6%) |
| Blood | 8 | 1 | 1 | 0 | 1 | 11(20.37%) |
| ETA | 11 | 0 | 0 | 0 | 1 | 12(22.22%) |
| Cathetre tip | 1 | 3 | 0 | 0 | 0 | 4(7.41%) |
| Sputum | 1 | 0 | 0 | 0 | 0 | 1(1.85%) |
| Wound swab | 2 | 0 | 0 | 0 | 1 | 3(5.55%) |
| Total | 32(59.26%) | 5 (9.26%) | 3(5.55%) | 3(5.55%) | 10(20.37%) | 54(100%) |

ICU: Intensive care unit, OPD: Outpatient clinic, ETA: Endotracheal aspiration.

**Table 4. Antimicrobial susceptibility testing for *Klebsiella pneumoniae*.**

| Antibiotics | Resistant | Intermediate | Sensitive |
|---|---|---|---|
| AMP | 54(100%) | 00(0%) | 00(0%) |
| AMC | 54(100%) | 00(0%) | 00(0%) |
| PTZ | 35(64.8%) | 3(5.55%) | 16(29.6%) |
| FOX | 26(48.1%) | 1(1.85%) | 27(50%) |
| CAZ | 45(83.33%) | 00(0%) | 9(16.66%) |
| CTX | 41(75.9%) | 00(0%) | 13(24.1%) |
| CPM | 40(70.4%) | 00(0%) | 14(25.9%) |
| ATM | 38(70.37%) | 00(0%) | 16(29.63%) |
| IMP | 20(37.0%) | 00(0%) | 34(62.96%) |
| MEM | 20(37.0%) | 00(0%) | 34(62.96%) |
| AN | 26(48.1%) | 00(0%) | 28(51.9%) |
| CM | 30(55.5%) | 11(1.85%) | 23(42.6%) |
| CIP | 34(62.96%) | 2(3.7%) | 18(33.33%) |
| TEM | 30(55.5%) | 0(0%) | 24(44.44%) |

Ampicillin (AMP), Amoxicillin-Clavulanic Acid (AMC), Piperacillin-tazobactam (PTZ), Cefoxitin (FOX), Ceftazidime (CAZ), Ceftriaxone (CRO), Cefotaxime (CTX), Cefepime (CFM), Aztronam (ATM), Imipenem (IPM), Meropenem (MEM), Amikacin (AN), Gentamicin (CN), Ciprofloxacin (CIP), Trimethoprim-Sulphomethoxazole (TMP).

The susceptibility results against 15 antimicrobial agents is shown in the "Table 4"and it was found that the most resistant drug were ampicillin followed by amoxiclave and cetftazidim.

In this study 54 isolates of *Klebsiella pneumoniae* were screened for ESBL production by using standard disk diffusion method [21]. The result showed that 40 (74.0%) isolates were resistance for 3rd generation of cephalosporins (ESBL producer), while 14 (26.0%) was susceptible for 3rd generation of cephalosporin (non ESBL producer).

All confirmed *Klebsiella pneumoniae* were tested for ESBL by double disk synergy method. The selected bacterial samples were tested against 3rd generation cephalosporin and characteristic key hole appearance were observed in 28 (51.85%) samples. At the same time all samples were tested by VITEK 2 compact system for ESBL and the same results were obtained. The rate of both methods was illustrated in Table 5 and statistically this relation is significant (*P* value < 0.05).

## Proportion of multidrug-resistance among isolates of *Klebsiella pneumoniae*

Multidrug resistance pattern were analyzed among all isolated *Klebsiella pneumonia* (Bora et al. 2014), any strain that shows resistant to at least one antibiotic in each group of three or more classes of antimicrobials reported as multidrug-resistance MDR [29]. Out of 54 samples of isolated *Klebsiella pneumoniae*; 32 (59.29%) were MDR "Table 6", and among all MDR; 75% were ESBL positive "Table 7".

**Table 5. Percentage of ESBL by screen and confirmation test.**

| ESBL test | ESBL screen positive | ESBL screen negative | Total |
|---|---|---|---|
| ESBL confirmation positive | 28 | 0 | 28 |
| ESBL confirmation negative | 12 | 14 | 26 |
| Total | 40 | 14 | 54 |

**Table 6. Percentage of MDR among *Klebsiella pneumoniae*.**

| Resistant profile | *Klebsiella pneumoniae number = 54* |
|---|---|
| Resistant to one group | 11 (20.37%) |
| Resistant to two groups | 11 (20.37%) |
| Resistant to three groups | 15 (27.77%) |
| Resistant to four groups | 17 (31.48%) |
| MDR (multi drug resistant) | 32 (59.25%) |

**Table 7. Relation of MDR and ESBL.**

| ESBL/MDR | MDR | Non-MDR | Toatal |
|---|---|---|---|
| ESBL positive | 21(75%) | 7(25%) | 28(100%) |
| ESBL negative | 11(42.3%) | 15(57.7%) | 26(100%) |
| Total | 32(59.26%) | 22(40.74%) | 54(100%) |

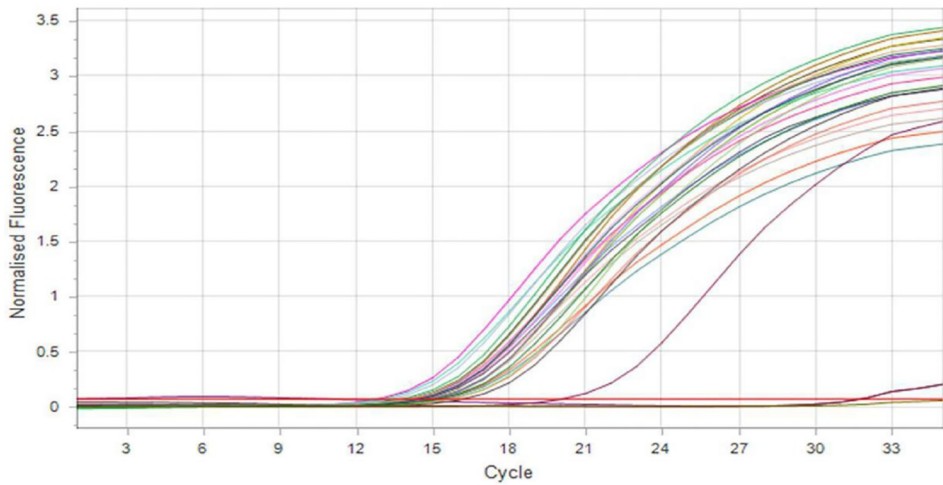

**Fig 3. SYBR green real-time PCR amplification result of *bla*SHV gene after 35 cycle (fluorescence versus cycle number).**

## Molecular detection of ESBL

Both ESBL genes (*bla* EM and *bla* SHV) were selected to be tested in this study by using real-time PCR. The result demonstrated that 26 isolates (92.85%) were positive for *bla*SHV gene (Figs 3 and 4) and 15 isolates (53.57%) were positive for *bla*TEM gene (Fig 5), and 15 isolates (53.57%) have both *bla*SHV and *bla*TEM genes, and 2 isolates (3.70%) out of ESBL positive isolates found to be negative to both genes.

## Discussion

*Klebsiella pneumoniae* is one of the most important multidrug resistant pathogenic bacteria that cause several diseases such as, burns infections, urinary and respiratory tract infections, wounds and blood infections and liver abscess.

All isolated *Klebsiella pneumoniae* in this study were from different wards in hospitals where different samples were analyzed and received from different clinical condition. 42.6% from all samples were urine specimen, this result is in agreement with study done in Erbil-

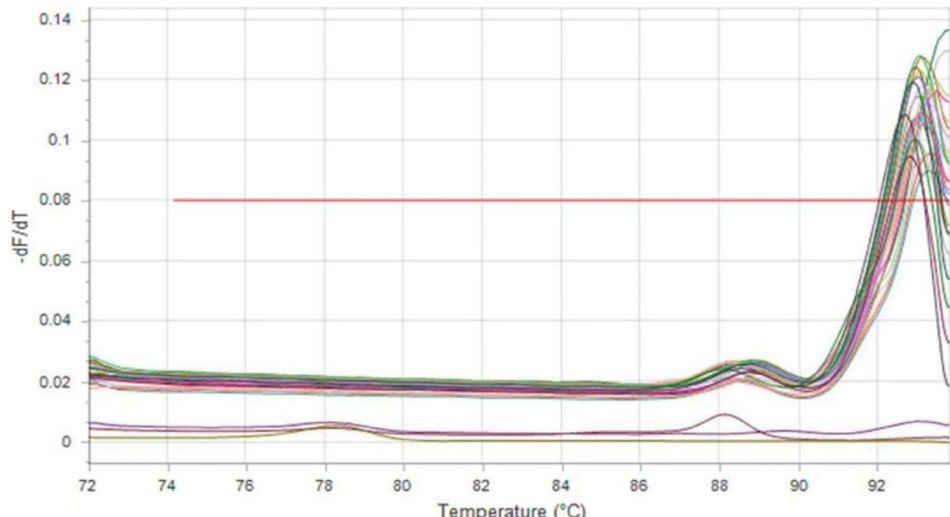

**Fig 4. Melting curve (fluorescence versus temperature) of *bla*SHV gene amplification products.**

Iraq; whom reporting higher percentage of urine sample 71.7%, this indicate that urinary tract infection is the most common infection in hospitals and Klebsiella is the second most common isolated bacteria after *E. coli* in this condition [31, 33]. The likelihood of *Klebsiella pneumoniae* hospital-acquired infections is greatly increased by the presence of invasive devices, such as catheters and ventilator associated pneumoniae in hospitalized patients and risk of infection increase with those patients with neuropathy bladder and diabetes mellitus [13].

The second most common specimen was blood (20.37%), which accounts for more than 70% and most of were of intensive care unit that indicates presence of more critical patients in this unite and they need blood culture continuously to exclude sepsis by gram negative bacteria. *Klebsiella pneumoniae* can persist on environmental surfaces human skin, within the respiratory, urinary tracts and easily transferred among patients via clinical operations and examinations, and it has become one of the most frequent outbreaks in intensive care Units [34].

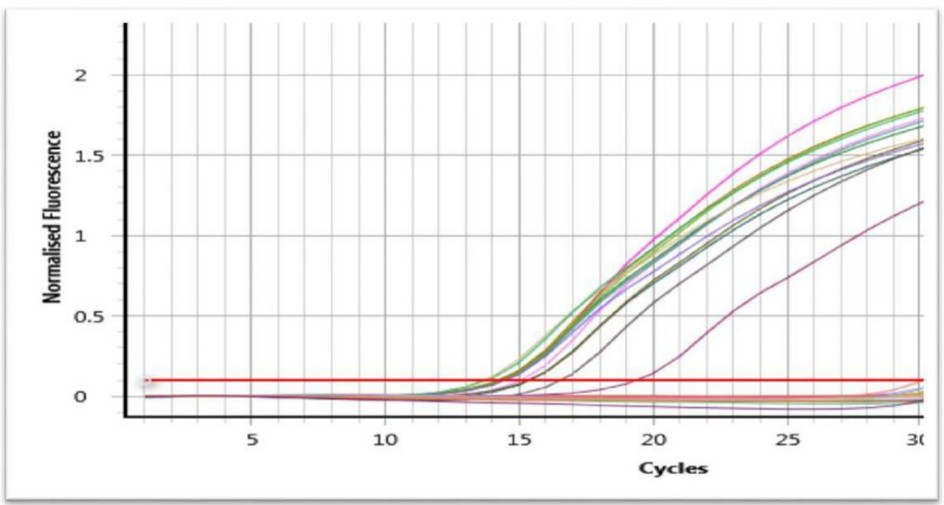

**Fig 5. SYBR green real-time PCR amplification result of *bla*TEM gene after 30 cycle (fluorescence versus cycle number).**

Antibiotic profile were tested against all isolated species and it was clear that 100% resistance to ampicillin was recorded which is in agreement to the studies performed in Erbil/ Iraq and Egypt [35, 36]. However; this high level of resistance to ampicillin is not surprising due to constitutive expression of a chromosomally encoded B-lactamase (*bla*SHV-1) which confers resistance to ampicillin, amoxicillin, carbenicillin and ticarcillin [37].

The other drug such as amoxicillin-clavulanic acid also shows high resistance rate (96.29%), the same result was reported in Erbil, Najaf and Duhok / Iraq, [38, 39] and similar results were reported in Nigeria, Iran, Saudi Arabia [40–42]. while lower resistance rate were recorded in a study done in Erbil-Kurdistan region-Iraq and Baghdad-Iraq [8, 43].

Regarding beta-lactam/beta-lactamase inhibitor combinations; the proportion of isolates showing resistance to amoxicillin/clavulanate (96.29%) was significantly higher than resistance to piperacillin/tazobactam (51.85%), the same result was recorded in Kingdom of Saudi Arabia [44]. The fact that piperacillin/tazobactam is less commonly used in this locality and in comparison to ampicillin which indicate over used of antibiotic causing selective pressure on bacteria for development of resistance.

The proportions of *Klebsiella pneumoniae* isolates showing resistance to non-beta-lactam agents such as trimethoprim (57.40%), ciprofloxacin (55.55%), gentamicin (53.70%), and amikacin (40.74%) were recorded, There is a bit difference with the result to what was recorded in a study done in Erbil-Sulaimani-Duhok/ Iraq by Khalid et al. [45], while imipenem and meropenem were the most susceptible antibiotics (63.0%) against *Klebsiella pneumoniae* isolates followed by Amikacin (59.26%), these results are in agreement with result of studies done in Erbil/Iraq and Iran [31, 41]. which stated that imipenem and meropenem were the most effective antibiotics, this result probably due to less frequent use of carbapenem to treat infection in compared to other antibiotics in this country. However, the emergence of resistance to carbapenem requires careful monitoring, as many of the producers are resistant to all antimicrobials posing a serious threat to hospital units, especially among immune compromise patients and among hospitalized patients. This finding was in agreement with data reported from other regional as well as global studies [31, 40, 46, 47] which may imply that increased use of carbapenems could potentially further select resistant strains.

The prevalence rate of ESBL was 51.85% among isolated *Klebsiella pneumoniae* by double disk diffusion method and VITEK 2 compact system, this result is in consistence with the study carried out in Erbil, Duhokin Iraq and in Iran [31, 39, 48], while higher rate was reported in Basrah-Iraq [49] and a lower rate was reported in a study that performed in Iran [50]. There are several studies from Turkey, Iran, Spain and Taiwan; recording prevalence rate of ESBL near to what was recorded in this study [41, 51, 52], this difference may be due to sample size type of clinical specimen. This variation in prevalence rate of ESBL may be due to differences in sample size.

The accurate detection of ESBL producing microorganisms is a major clinical problem in the laboratories, requiring not only phenotypic tests but also genotypic tests for all genes associated with β-lactamase production. The prevalence of β-lactamase producers and the distribution of ESBL genotypes are different from one year to another and even vary greatly in different geographical areas [53].

ESBL enzyme carries by several genes such as; SHV, TEM which were analyzed in this study, both genes (*bla*SHV and *bla*TEM) were detected by real-time PCR using specific primer sequences which yielded product sizes of 450 bp and 296 bp, respectively.

The result demonstrated that out of 28 ESBL positive strains 26 (92.85%) of them were positive for *bla*SHV while15 (53.57%) of them were positive for *bla*TEM. This means that 92.85% of ESBL producer *Klebsiella pneumoniae* were carried at least one of the both genes; similar result was reported in a study done in Erbil, Sulaimani and Duhok in Iraq by Khalid et al. [45].

This result was in agreement with several studies [36, 39, 52], whom they reported the prevalence of *bla*SHV gene in *Klebsiella pneumoniae* isolates near to the prevalence rate in this study but TEM gene were recorded a bit higher rate than this study.

Out of all ESBL positive isolates by phenotypic methods; 2 isolates showed negative result by genotyping; this may be due to presence of other ESBL genes such CTX-M, and OXA gene.

Resistance pattern among all isolated *Klebsiella* were analyzed and MDR strain were found in 59.25%; this indicate that not all samples carry MDR phenomenon. The remaining negative isolated may carry other genes such as AmpC, KPC, or other type carbapenemase enzyme but among all ESBL producers 75% were MDR, similar result were reported in a study in Erbil/Iraq [2] and India [54]. This mostly due to the presence of multiple resistance gene on the same cassette of chromosomal, The plasmids that harbor ESBL genes also carry resistance genes for other antibiotic classes such as aminoglycosides, chloramphenicol, sulfonamides, trimethoprim, and tetracycline. Thus, Gram negative bacilli containing these plasmids mostly are multidrug-resistant bacteria [4]. Multi drug resistance *K. pneumoniae* strains have caused many disease problems worldwide with higher morbidity and mortality rates, posing a considerable threat to public health [38].

## Conclusion

The prevalence and spread of ESBL-producing *Klebsiella pneumoniae* strains is worrisome and it is a frequent isolates among different clinical samples from different conditions, specifically from intensive care unit, both ESBL genes were isolated among Klebsiella Pneumoniae with prevalent of *bla*SHV over *bla*TEM gene.

## Supporting information

**S1 Data.**
(XLSX)

## Acknowledgments

Particular thanks to Ministry of Health and General Health Directorate of Sulaimani with Department of Laboratory for granting to this work and special thanks to laboratory staff of microbiology department in Shar Teaching Hospital.

## Author Contributions

**Formal analysis:** Aso Bakr Mohammed.

**Funding acquisition:** Khanda Abdullateef Anwar.

**Methodology:** Aso Bakr Mohammed.

**Project administration:** Khanda Abdullateef Anwar.

**Software:** Aso Bakr Mohammed.

**Supervision:** Khanda Abdullateef Anwar.

**Validation:** Khanda Abdullateef Anwar.

**Writing – original draft:** Aso Bakr Mohammed.

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
