## [Decision Letter · Decision Letter 0]

15 Nov 2021

PONE-D-21-32646Phenotypic and genotypic detection of extended spectrum beta lactamase enzyme in Klebsiella PneumoniaePLOS ONE

Dear Dr. Anoar,

Thank you for submitting your manuscript to PLOS ONE. After careful consideration, we feel that it has merit but does not fully meet PLOS ONE’s publication criteria as it currently stands. Therefore, we invite you to submit a revised version of the manuscript that addresses the points raised during the review process.

Please address the reviewers recommendations and prior to submission make sure that the manuscript has been well revised by several people. In its current form it has gramatical and spelling errors and the names of the bacteria contains spelling mistakes and it is not italics. Please, pay attention to the details. ==============================

We look forward to receiving your revised manuscript.

Kind regards,

Monica Cartelle Gestal, PhD

Academic Editor

PLOS ONE

Journal Requirements:

Reviewers' comments:

Reviewer's Responses to Questions

**Comments to the Author**

1. Is the manuscript technically sound, and do the data support the conclusions?

Reviewer #1: Yes

Reviewer #2: Partly

2. Has the statistical analysis been performed appropriately and rigorously? 

Reviewer #1: No

Reviewer #2: Yes

3. Have the authors made all data underlying the findings in their manuscript fully available?

Reviewer #1: Yes

Reviewer #2: Yes

4. Is the manuscript presented in an intelligible fashion and written in standard English?

Reviewer #1: No

Reviewer #2: No

5. Review Comments to the Author

Reviewer #1: N0 Line number Reviewer’s Comments/Suggestions

1 Abstract Most results of antibiotic not clarify clearly in abstract section

Keywords Need to arrange alphbetically

2 The manuscript contains many grammatical errors that must be corrected especialy the discussion and introduction section need too much correction

3 highlighted in Green Correct the word

4 highlighted in Red Delete word or character

5 Many Word not writed in fully form like” EBLS,ICU,U.S.A,MICs,OPD)and others, also the term EBLS sometimes writed “ESBLS” other palces writed “ESBLs” it must write in unique and scinetific style

6 Highlighted in Yellow Give a space

7 Results - İn this sectio the concentration of each antibiotiv must mentioned inside the (table 4).

- (P-value with 0.00001) this wrong value (correct the value).

8 Discussion The authors mentioned in this section this wrong reference lined with yellow” The other drug such as amoxicillin-clavulanic acid also shows high resistance rate (96.29%), the same result was reported in Erbil , Najaf and Duhok / Iraq (Gupta etal.(2003), (Aljanaby etal.(2018), (Qadir (2019). Respectively. So Gupta and his collegous study founded in U.S.A

9 References All the refernce not in one unique style; the year of publication sometime after the name of authors and sometimes after the title of article, sometiems after the year of publication come comma others semicolon

10 Others important notice - Many word wtrited wrongly like „standardise,Fifty four,30 µg must become

30µg (without any space),TEM and SHV writed with different format like balSHV,blasCTM

- Table 1,2 need to re-type in word format not in picture foramt

- Chi-sequare and correlation analysis test mentioned in statistical anlalysis section but actualy not calculate for the data (for comparision), the authors must delete these characters , also the authors must mentioned which version of M.soft excel they used

- Table 3, 4,5 ,6 and 7 must re-type again with word format adn without coluration .

- The manuscript not contanin conclusion and any Acknowledgement.

Reviewer #2: After reading the manuscript titled: I suggest the need for major revisions.

I believe that the presented research is very relevant and worth publishing, however the authors need to pay attention to a few issues detailed below. First of all, spelling inconsistencies, such as VITEK / Vitek, are noticeable. The authors do not pay attention to the correct spelling of the name of the microorganisms (no italics, no capital letters), the names of the authors cited are often written in lower case. Punctuation errors are also noticeable.

Introduction:

1) “Extended-spectrum beta-lactamase (ESBL)-producing bacteria is plasmid-mediated enzymes…” – should be corrected, bacteria aren’t enzymes!

2) “…evolutionary families based on their amino acid sequence TEM and sulphydryl variable SHV were the major types…” is this one sentence?

3) ” The fact that patients most likely to become infected with ESBL-producing Enterobacteriaceae are those…” – Enterobacteriaceae should be in italics

4) “It is estimated that Klebsiella species cause 8% of all hospital-acquired infections...” – Klebsiella should be in italics

5) “Antimicrobial resistance in Klebsiella Pneumoniae…” – pneumoniae should be lowercase

6) “Combinations of all types of bla genes were reported in this species, this may be due to either carriage of an antibiotic-resistant plasmid encoding an array of antibiotic resistance genes or due to acquisition of transposons containing different bla genes on the same plasmid” – bla should be in italics

7) “Many different techniques exist for confirming ESBL production , Phenotypic tests for ESBL detection” – this sentences should be separated by a period, not a comma. Do not put spaces in front of the punctuation marks

8) “(Schwarz etal.(2010)” – too much brackets

9) “(shariif etal.(2017).” – too much brackets, write surnames with capital letters

10) “Another confirming method involves at least an eight-fold reduction in the MICs of ceftazidime or cefotaxime in the presence of clavulanic acid, The isolates then reported as resistant to all penicillins, cephalosporins and aztreonam (muhamed & Abas 2019).” - Separate sentences with a period and not with a comma, and write surnames with a capital letter

11) “(philippon etal.(2011).” – surnames with a capital letter

Methods

1) shon etal.(2013). – surnames with a capital letter

2) biomeireux – company name should be write: bioMérieux

3) “Vitek® 2 biomeireux machine was used for the identification of all isolated Klebsiella according to the manufacturer’s recommendations” – which card was used?

4) tryptic soya broth - who is the medium’s producer?

5) Eppendoruf – correct to Eppendorf

6) “Antibiotic susceptibility pattern of Klebsiella pneumoniae isolates were performed according to the Kirby–Bauer disk diffusion method (Bauer etal.(1966) by using 15 commonly used antibiotic agents and all of the inoculated plates were aerobically incubated at 37°C for 18–24 hours, and the results were interpreted based on the instruction provided by the CLSI (2018 ).” - please mention here the antibiotics and their concentration

7) “disks” – correct to “discs” in each place

8) “Mueller-Hinton Agar plate” – no information about producer

9) VITEK or Vitek – please standardize

10) “Then let it to dissolve well and leave it for 15 minute at room temperature and mixed by vortexing, stock primer diluted 1:10 (10 μL of stock primer with 90 μL of DNA and RNA free distil water) to prepare working primer in concentration 10 pmol/μL (Ali& Ismael (2017).” - please correct the syntax so that it doesn't look like an instruction

Discussion

1) “this indicate that UTI is the most common infection in hospitals and Klebsiella is the second most common isolated bacteria” – Klebsiella should be in italics

2) “ICU samples . this indicates” – remove space before period and start sentence with a capital letter

6. PLOS authors have the option to publish the peer review history of their article (what does this mean?). If published, this will include your full peer review and any attached files.

Reviewer #1: No

Reviewer #2: No

---

## [Author Response · Author response to Decision Letter 0]

14 Dec 2021

1/ all Klebsiella pneumoniae were checked.

2/ tables written and putted in word format 

3/ figure uploaded in APACE style 

4/ Reference checked according to submission guide line 

5/ all reviewer comment checked step by step 

6/ 2 file uploaded one is manuscript ( re new one after correction ) , other revised one as follow : green word : corrected and newly added according to reviewer comment , red words : deleted word 

7/ cover page checked according to file template of plos one \\

---

## [Decision Letter · Decision Letter 1]

5 Apr 2022

Phenotypic and genotypic detection of extended spectrum beta lactamase enzyme in Klebsiella Pneumoniae

PONE-D-21-32646R1

Dear Dr. Anoar,

We’re pleased to inform you that your manuscript has been judged scientifically suitable for publication and will be formally accepted for publication once it meets all outstanding technical requirements.

Kind regards,

Monica Cartelle Gestal, PhD

Academic Editor

PLOS ONE

Additional Editor Comments (optional):

Reviewers' comments:

Reviewer's Responses to Questions

**Comments to the Author**

1. If the authors have adequately addressed your comments raised in a previous round of review and you feel that this manuscript is now acceptable for publication, you may indicate that here to bypass the “Comments to the Author” section, enter your conflict of interest statement in the “Confidential to Editor” section, and submit your "Accept" recommendation.

Reviewer #1: All comments have been addressed

Reviewer #2: All comments have been addressed

2. Is the manuscript technically sound, and do the data support the conclusions?

Reviewer #1: Yes

Reviewer #2: Yes

3. Has the statistical analysis been performed appropriately and rigorously? 

Reviewer #1: Yes

Reviewer #2: Yes

4. Have the authors made all data underlying the findings in their manuscript fully available?

Reviewer #1: Yes

Reviewer #2: Yes

5. Is the manuscript presented in an intelligible fashion and written in standard English?

Reviewer #1: Yes

Reviewer #2: Yes

6. Review Comments to the Author

Reviewer #1: (No Response)

Reviewer #2: (No Response)

7. PLOS authors have the option to publish the peer review history of their article (what does this mean?). If published, this will include your full peer review and any attached files.

Reviewer #1: **Yes: **JALEEL SAMANJE

Reviewer #2: No

---

## [Editor Report · Acceptance letter]

7 Apr 2022

PONE-D-21-32646R1 

Phenotypic and genotypic detection of extended spectrum beta lactamase enzyme in *Klebsiella pneumoniae*

Dear Dr. Anwar:

I'm pleased to inform you that your manuscript has been deemed suitable for publication in PLOS ONE. Congratulations! Your manuscript is now with our production department. 

Kind regards, 

on behalf of

Dr. Monica Cartelle Gestal 

Academic Editor

PLOS ONE